# A systematic review and meta-analysis of studies testing effects of cash transfers on child cognitive, language, and socio-emotional development in low- or middle income countries

Lia C. H. Fernald [1] ✉, Eleanor Tsai [2] & Paul J. Gertler[3]

Cash transfers alleviate financial hardship as either conditional programs, requiring behavioral compliance, or unconditional programs. We undertook a systematic review, quality appraisal, and meta-analysis of 16 RCTs following PRISMA guidelines, examining effects of cash transfers on child development among children ($n = 29,887$, <8 years old) in low- or middle-income countries; papers were excluded if they used quasi-experimental or simulation methods, or did not have a control group without cash. To examine risk of bias in the studies, we used the revised Cochrane risk-of-bias tools for individually randomized trials (RoB 2) and cluster randomized trials (RoB 2 CRT). After 3.6 years, cash transfers had a small but significant positive effect on cognitive (Cohen's $d = 0.08$, 95% CI 0.04, 0.13), language ($d = 0.09$, 95% CI 0.04, 0.13), and gross motor outcomes ($d = 0.07$, 95% CI 0.03, 0.11); only conditional cash transfers showed significant effects on socio-emotional outcomes ($d = 0.17$, 95% CI 0.04, 0.31). Across outcomes, findings were strongest for children in families receiving conditional cash transfers and a sub-set of cash transfer programs ("cash-plus", which integrated health, nutrition, or parenting support). Limitations of the analysis are that we could only include a small number of studies, and that there was a great degree of heterogeneity among cash transfer programs, and thus our findings are exploratory. In spite of these limitations, our findings suggest that cash transfers were most effective to promote child development when the transfers were bundled with crucial services or support for families, not simply as cash alone.

Roughly one billion children – almost half the world's children – experience multi-dimensional poverty, which means they do not have access to the healthcare, nutrition, shelter, or education that they need to survive and thrive[1]. Poverty remains a primary driver of adverse health outcomes and negatively impacts children's development, due to the large number of risk factors associated with living in impoverished environments[2]. Poverty interacts with various socio-cultural, psychosocial, and biological risk factors to influence child development, ultimately affecting long-term adult productivity[3–5]. Identifying and implementing programs that effectively and

sustainably improve child development has been a challenge for policy-makers, particularly in low- or middle-income countries[6].

Poverty has direct and long-lasting effects on child health and development, which have implications for concurrent well-being as well as for long-run welfare[2]. Early cognitive and language skills can affect academic achievement, educational attainment, earnings, and economic potential[7]. Child development is dynamic and results from repeated interactions of biological factors (genes, brain growth, neuro-muscular maturation) and environmental influences (parent-child relationships, community

[1]Division of Community Health Sciences, School of Public Health, University of California, Berkeley, CA, USA. [2]Division of Health Policy and Management, School of Public Health, University of California, Berkeley, CA, USA. [3]Haas School of Business, University of California, Berkeley, CA, USA. ✉e-mail: fernald@berkeley.edu

characteristics, cultural norms) over time[8]. For example, children can only learn to walk if there is a convergence of physiologic systems such as muscles, social-emotional motivation to move independently, and adequate support from caregivers to practice the emerging skill[9–11]. Similarly, children benefit from schooling only to the extent that they are school-ready in terms of literacy, numeracy, and self-control[12]. Children who are not school-ready may benefit less from school, and may fall behind other school-ready children; this inequality can then be a major source of poverty and poor outcomes later in life[13].

The foundation for cognitive development is established very early in life, as core neurodevelopmental processes such as neuron proliferation, axon and dendrite growth, and synaptogenesis begin during gestation and continue throughout infancy[2]. Development throughout the lifespan proceeds in trajectories with one phase laying the groundwork for development at the next phase. The progression of cognitive and language abilities includes acquiring capacities such as sustaining attention, understanding and following directions, working independently, and problem-solving, as well as socio-emotional skills, such as getting along with others, managing, and expressing emotions. The development of these skills is a recursive process in which a child's level of skills at any time point affects how much they will benefit from investments in those skills, a feature known as dynamic complementarity. In this way, learning compounds over time, and the earlier that a child can master foundational capacities, the more efficient their subsequent schooling and educational opportunities will be. Program inputs such as nutritional supplementation or psychosocial stimulation can also compensate for deficits in the household environment, making them a vital mechanism for promoting equity.

Young children can be supported by a range of programmatic options, including nutrition supplementation, improved water, sanitation, and hygiene, or investments in schooling and education[14]. One approach to promoting optimal child outcomes is through the provision of safety net programs, which support vulnerable populations by distributing transfers to low-income households to prevent shocks, protect the chronically poor, promote capabilities and opportunities for vulnerable households, and transform systems of power that exclude certain marginalized groups (for example, women and children)[15]. Cash transfer programs are designed to mitigate the immediate consequences of poverty and are a widely used and popular approach to provide safety net support and financial assistance to recipients without behavioral prerequisites, granting families full autonomy over how the funds are utilized. A foundational premise of cash transfers is that parents are income-constrained and do not have money to afford their families' most pressing needs, such as nutritious food or medical treatment. By expanding a family's purchasing power, caregivers can choose what goods they want to buy and make choices about the amount and quality of their purchases. A recent World Bank report identified 203 countries reporting 962 different cash transfer programs, the majority of which had been introduced during the pandemic when 17% of the world's population (1.36 billion people) received some form of cash transfers[16].

Cash transfers are classified as conditional when their receipt hinges on strict behavioral compliance of families with a set of requirements (the "conditions" of the cash transfer) usually related to health or education[17]. Frequently, the requirements relate to preventive healthcare services or health and nutrition education sessions designed to promote positive behavioral changes, and many programs also require school attendance for school-age children. The direct conditions of conditional cash transfer programs offer benefits, for example requiring a pregnant women to obtain prenatal care or parents to take their children to growth monitoring and vaccination appointments. In additiona, caregivers may be required to attend health education programs, which may also play a role in increasing knowledge relating to appropriate child rearing. Ultimately, translating healthcare utilization or health education workshop attendance into improved health outcomes depends directly on the quality and delivery of healthcare services and education programs.

One of the key mechanisms by which cash transfer programs can improve child development is through the family investment model[18,19].

With the influx of additional financial resources via the cash transfer, caregivers are better positioned to enhance the home environment (such as upgrading flooring, roofing, or water access), through the purchase of goods that influence child growth and development (e.g. more expensive or nutritious foods, healthcare, books, or toys), or by spending more time with children[20]. When the goal is to improve child outcomes, conditional cash transfer programs usually target transfers to women based on evidence that money controlled by mothers, in contrast to fathers, is spent on more child-centric goods and services[21,22]. Although some nations offer free public healthcare, cash transfers provide extra income, which could be used to pay for transportation to and from health centers, or could allow families to pay for medical treatments that may not be provided for free.

Another possible mechanism by which increased cash transfer programs could be linked to improvements in child development is via the family stress model, which connects greater household resources to reduced maternal depression[23], and fits into a broader framework of the psychology of poverty[24]. In this framework, additional income could indirectly improve the psychological well-being of family members through reductions in subjective feelings of financial strain and material deprivation, which could then be associated with improvements in child well-being and achievement[25]. Severe economic pressure on a household is known to compromise parental mental health and also to degrade intra-familial relationships[26]. Children of depressed mothers are at a higher risk for reductions in mental and motor performance, disturbances in early attachment relationships, increased rates of behavioral problems[27,28], and adverse child outcomes[29], primarily due to inadequate childcare practices[30]. Furthermore, maternal psychopathology acts as a mediating pathway connecting poverty or other stressors (such as exposure to domestic violence or war) and children's behavior problems[31]. Since poverty affects the ways in which parents monitor their children, provide stimulation for their children, and respond to their needs[32], increasing access to economic resources may allow parents to be more responsive, warm, and consistent[33].

Families receiving financial transfers tend to prioritize spending on food security and productive investments such as livestock ownership and agricultural inputs rather than temptation goods[34]. Additionally, cash transfer program participation has been linked with significant reductions in mortality in women and in children under five[35], improvements in subjective well-being and mental health among adults[36], as well as benefits to several population-wide health outcomes such as antenatal care, facility deliveries, and some nutrition outcomes[37]. In children, cash transfer programs are beneficial for growth[38], as well as for anemia[39], and school attendance[40], with some evidence that cash paired with other intervention components such as assets, training packages, and messaging ("cash-plus") outperform cash-only programs for child health outcomes[41]. Interventions that deliberately combine cash with parenting training, psychosocial stimulation, education on infant and young child feeding practices, nutritional supplementation or counseling, growth monitoring, or other maternal and child health messaging could be especially influential for early child development outcomes. The evidence to date, however, regarding the effects of cash transfer programs on child development outcomes (e.g., cognition, language, motor, socio-emotional) has been mixed.

In a recent book chapter, we provide a conceptual review of the theoretical pathways linking poverty, social safety nets, and child outcomes, synthesizing broader evidence related to early child development, physical growth, nutrition, and general health outcomes[42]. In this paper, our primary goal was to add to the existing literature about cash transfer programs by examining their specific effects on early child development. We focused on randomized controlled trials specifically examining four domains: cognition, language, socio-emotional, motor (gross and fine) skills, updating our prior review[43] with new evidence. We also examined heterogeneity of findings by whether the transfer was conditional or not, whether the program was cash-plus or cash only, and the size of the transfer. Our hypotheses were that cash transfer programs would have small but significant effects across all child outcomes, and that effects would be concentrated in the children of households receiving conditional cash transfers.

## Methods
### Search strategy
We began with our previously published review examining cash transfers and child development outcomes[43,44], and then searched PubMed, EconLit, PsycInfo, references from previous review papers, and Google Scholar for additional gray literature papers using keywords corresponding to the outcome domains of early child development. We conducted an initial search on March 6, 2025, and a second search on September 15, 2025, focusing on papers that had been published after the previous reviews. For program type, the search terms included "cash transfer" or "cash-plus". Outcomes included "child" and ("development", "language", "cognition", "motor", "behavior", "socio-emotional", or "executive function"). Search terms for study design included: "RCT" or "randomized". We adhered to the World Bank classifications of countries for the years in which the studies were conducted and included studies from both low- and middle-income contexts.

### Study inclusion
All papers returned from these queries were screened by two reviewers (LF and ET) for relevance and experimental methodology based on title and abstract, and included for full data extraction after consensus among all three authors on meeting the inclusion criteria. The review process was compliant with PRISMA guidelines to ensure accuracy and minimize selection bias. The final list of studies included in the review was restricted to studies that (i) measured early childhood development outcomes for children ages 0–8, (ii) were either individually randomized or cluster randomized, and (iii) evaluated cash-only or cash-plus treatments relative to a control group without cash (Fig. 1). Thus, papers were excluded if they did not measure outcomes in any of the five early childhood development domains; if they were reviews or used quasi-experimental or simulation methods; or if they did not have a control group without cash. When a study had multiple eligible treatment arms delivering cash transfers (or cash transfers and a plus component), the effect sizes for each treatment arm were reported individually.

### Study quality assessment
We reviewed evidence from studies examining the effects of conditional or unconditional cash transfer programs on cognitive, language, socio-emotional development, or motor development. To examine risk of bias in the studies, one reviewer (ET) used the Revised Cochrane risk of bias tools for individually randomized trials (RoB 2) and cluster randomized trials (RoB 2 CRT), which use a series of signaling questions to assess bias in five domains: (i) bias arising from the randomization process; (ii) bias due to deviations from intended interventions; (iii) bias due to missing outcome data; (iv) bias in measurement of the outcome; (v) bias in selection of the reported result[45,46].

### Estimating effects of cash transfer programs
Using standard techniques, we examined program effect sizes in aggregate, and also by the type of program: unconditional, unconditional-plus, conditional, or conditional-plus, where plus denotes a "cash-plus" program that combines cash transfers with one or more additional intervention components, compared to a "cash-only" program that provides transfers but no other services or information. Standardized effect sizes and confidence intervals were used directly wherever reported. In the case where an unstandardized regression coefficient was reported, we calculated a standardized effect size by dividing by the standard deviation of the control group, and we calculated a confidence interval based on the standard error of the resulting effect size. If the standard deviation of the control group was not reported, it was approximated by the standard deviation of the full sample.

### Exploring heterogeneity in program effects
We plotted associations between transfer generosity and early childhood outcomes, with transfer generosity expressed as either: (1) a percentage of household income or expenditure as reported directly in the study; (2) a percentage of per capita income or expenditure as reported directly in the study; (3) a percentage of household income or expenditure as reported in a previously published study of the same program; or (4) in the absence of the above information, approximated by the value of the transfer as a percentage of the product of per capita expenditures and average household size reported in the study. Studies that did not report any of the above information were excluded from this analysis. Given the heterogeneity in study populations represented and reported information, these plots should be considered heuristic in nature rather than as attempts to quantify the association between transfer generosity and effect size. To examine heterogeneity visually, we prepared violin plots to examine findings by program conditionality and region of study. Given the small number of papers included in the study and the lack of sufficient variation in program conditionality within regions, we were not able to run statistical tests examining whether the heterogeneity was significant. All analyses were completed using the *meta* and *metan* packages in STATA version 17.0. The protocol was registered post data extraction and synthesis via OSF (https://osf.io/2wbfg).

## Results
### Description of programs included in analysis
Our search yielded 208 records, which were reduced to 180 after removing duplicates (Fig. 1). The sample was then narrowed to 16, due to missing early child development outcomes ($n = 125$), insufficiently rigorous study design ($n = 18$), children out of age range ($n = 11$), and other reasons. The studies included in our analysis came from several regions around the world, including Africa (Burkina Faso[47], Madagascar[48], Niger[49], Nigeria[50], Tanzania[51], Uganda[52]), Asia (Bangladesh[53], China[54], India[55], Nepal[56]), and Latin America (Ecuador[57,58], Honduras[59], Mexico[60,61], Nicaragua[62]). The programs in each country varied in terms of cash amount, conditionality, and target for the program (Table 1), with most programs in Africa and Asia being unconditional and most programs in Latin America being conditional. All papers reported at least one measure of language, cognition, socio-emotional outcomes, and/or gross/fine motor development (Table 2). Cash-plus programs were highly variable and could include parenting support, nutrition counseling, or other maternal or child health messaging as the "plus" component (Table 3).

### Assessment of bias
We found that overall, half of the cluster randomized controlled trials had low risk and the remaining half had some concerns across all domains, using RoB 2 for individually randomized trials and cluster RCTs (Fig. 2). The potential for bias was greatest in the domain of deviation from intended interventions, with "some concerns" for 57.1% of the studies, and the domain of measurement of the outcome, with "some concerns" for half of the studies. Common concerns in these two domains were: participant awareness of their assigned intervention (cash or cash-plus), lack of information on blinding of (or inability to blind) assessors and intervention delivery agents to participants' assigned intervention, and lack of information on whether there were deviations from the intended intervention that arose from the trial context (i.e., recruitment and engagement activities that could have influenced receipt of cash or take-up of plus components, or any unconscious or conscious processes by trial personnel that could have undermined delivery of cash or plus components). Similarly, for the individually randomized trials, there were "some concerns" about deviations from intended interventions as participants were aware they were receiving a preschool voucher (China) or a cash transfer and/or center-based childcare (Uganda). The studies did not provide information on whether teachers were aware of students' assigned intervention, nor any potential deviations that could have arisen from the trial context.

### Effects of cash transfer programs on child outcomes
There were small, but positive effects of cash transfers across child development outcomes in children (Fig. 3), with a large degree of heterogeneity in

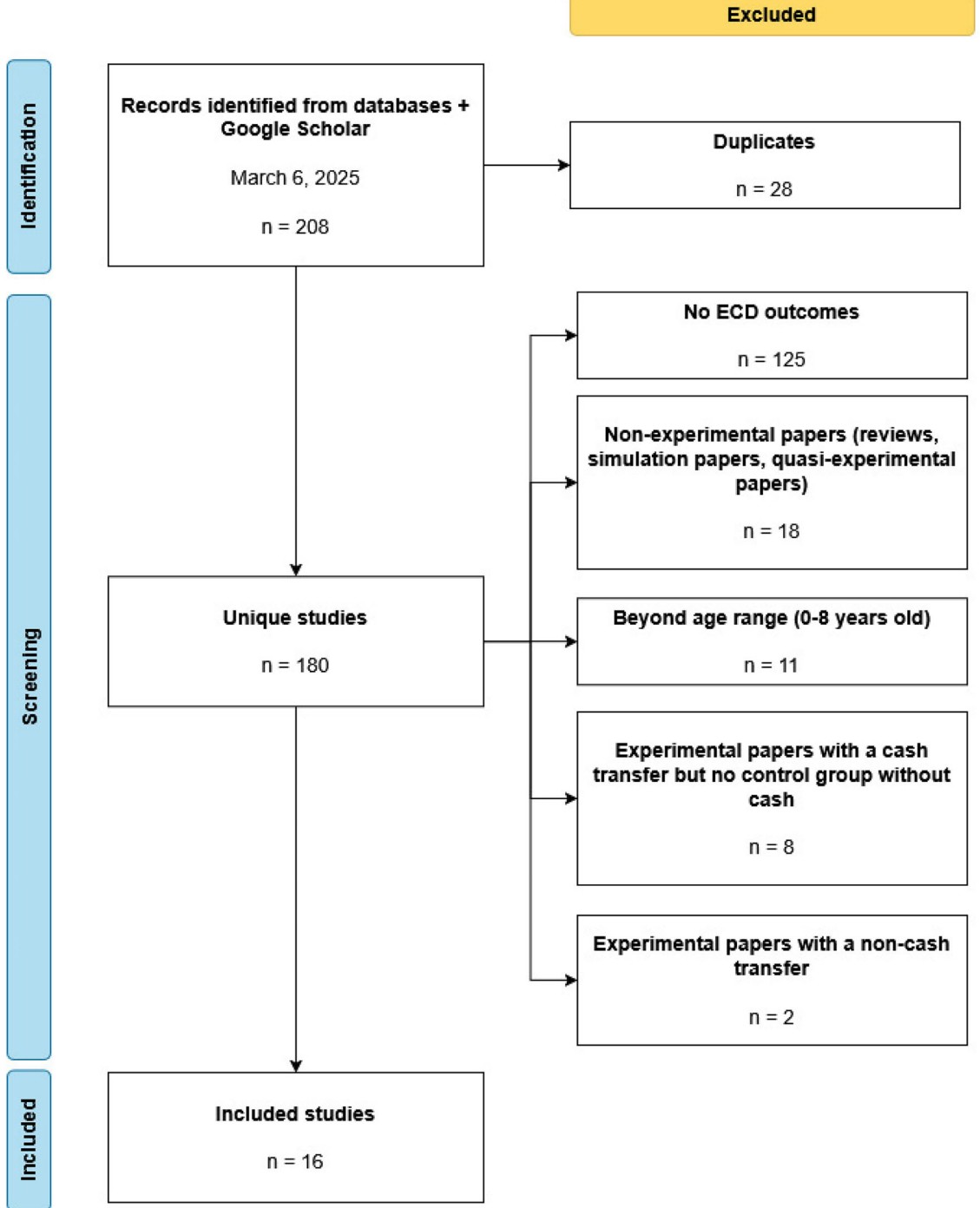

**Fig. 1 | PRISMA Diagram.** Flow diagram illustrating the number of studies included and excluded at each stage of the literature search (identification, screening, and final inclusion).

terms of effect sizes relating to program characteristics (Supplemental Fig. 1). For cognition, the studies reporting this outcome showed an average effect size of 0.08 (95%CI 0.04, 0.13) (Fig. 4). Two of the three conditional cash transfer programs showed impacts on cognition, with an effect size of 0.10 in Mexico and 0.52 in Honduras, but none of the estimates for effects of unconditional cash transfer programs (without a "plus" component) on cognition were different from zero. One unconditional cash-plus program (Bangladesh) achieved a cognitive effect size of 0.32 SD, with a "plus" component of psychosocial stimulation. The other unconditional cash-plus program (India) that produced significant impacts on cognition only achieved an effect size of 0.09 SD.

Effects of cash transfers on language showed an average effect size of 0.09 (0.04, 0.13) (Fig. 5). Similar to the findings for cognition, the three conditional cash transfer programs had the largest pooled effect size at 0.18 SD (0.13, 0.24), followed by the three conditional cash transfer-plus programs at 0.11 SD (0.04, 0.17). Only three of the twelve estimates from unconditional cash transfer or unconditional cash transfer-plus programs (Madagascar, Uganda, Bangladesh) showed significant effects on child language outcomes.

Cash transfer effects on socio-emotional outcomes in children were insignificant overall 0.04 (−0.01, 0.09), but were significant for conditional cash transfer programs with an effect size of 0.17 SD (0.04, 0.31), and

**Table 1 | Study Characteristics, including country, program type, transfer amount, age range and sample size**

| Study (listed alphabetically by country) | Program name (if any) | Type(s) of programs[a] | Cash only or cash-plus plus component(s)[a] | Transfer generosity (% of income/ expenditure) | Target child age range[b] | Sample size[c] |
|---|---|---|---|---|---|---|
| Bangladesh[53] | – | UCT/UCT+ | Arm II: Cash only Arm I: Psychosocial stimulation | 5% of monthly income | 6–16 months | 553 |
| Burkina Faso[47] | – | UCT/UCT+ | T1: Cash only T2: Livestock T3: Livestock, fortified flour, and nutrition information | 40% of the poverty line | In utero–5 years | 2200 |
| China[54] | – | CCT+ | Preschool tuition voucher | 20% annual rural per capita income | 4 years | 131 |
| Ecuador[57] | Bono de Desarrollo Humano | UCT | Cash only | 10% increase in family expenditure | 0–6 years | 1894 |
| Ecuador[58] | Bono de Desarrollo Humano | UCT | Cash only | 6–10% of household pre-transfer expenditure | 0–6 years | 1196 |
| Honduras[59] | Bono 10,000 | CCT | Health: Cash conditioned on health checkups for children ages 0–5 Education: Cash conditioned on school enrollment for children ages 6–18 | 13.5–18% of the median per capita consumption | 0–18 years | 1702 |
| India[55] | – | UCT+ | Nutrition messaging | 3.5% of the median per capita consumption | 0–2 years | 2163 |
| Madagascar[48] | Human Development Cash Transfer | UCT | Cash only | 30% of the average household consumption | 0–6 years | 3189 |
| Mexico[60] | Oportunidades | CCT | Cash only conditioned on health checkups for children 0–19 years old and school attendance for children in 3rd–9th grade | 25% of household income | 0–19 years | 2913 |
| Mexico[61] | Oportunidades | CCT | Cash only conditioned on health checkups for children 0–19 years old and school attendance for children in 3rd–9th grade | 25% of household income | 0–19 years | 945 |
| Nepal[56] | – | UCT+ | Local health worker-delivered information sessions on infant health and development | 8–20% of the median monthly income | In utero–2 years | 2614 |
| Nicaragua[62] | Atencion a Crisis | CCT+ | Group 1: Cash conditioned on school attendance for children ages 7–15 and nutrition, health, and education information package Group 2: Cash conditioned on school attendance for children ages 7–15 and vocational, labor market, and business skill training package Group 3: Cash conditioned on school attendance for children ages 7–15 and a lump-sum payment for a business development plan | 15–26% percent of per capita expenditure | 0–15 years | 3326 |
| Niger[49] | – | UCT | Optional health, nutrition, psychosocial stimulation, and child protection information package | 12% of total consumption | 6–59 months | 1602 |
| Nigeria[50] | Child Development Grant Programme | UCT+ | Maternal and child health information package | 12% of household monthly earnings | In utero–2 years | 3688 |
| Tanzania[51] | – | CCT+ | Community health worker-delivered health, nutrition, and responsive stimulation | 10% of the total monthly household income for subsistence farmers | 0–1 year | 405 |
| Uganda[52] | – | UCT/UCT+ | T2: Cash only T3: Center-based childcare | 32.4% of household income | 3–5 years | 1366 |
| **Total sample size** | | | | | | **29,887** |

[a]More than one type of program or plus component indicates a study with multiple treatment arms.
[b]Based on program eligibility criteria or transfer conditions (if any). If not specified, the child age range at baseline is reported.
[c]Total sample size across all treatment arms. If sample sizes are reported by outcome, the sample size for cognition is reported.

**Table 2 | Study Instruments Used to Measure ECD Outcomes listed by program**

| Study (listed alphabetically by country) | Child age at endline | Instruments used by the ECD domain | | | | Instruments validated for the study population? |
|---|---|---|---|---|---|---|
| | | Language | Cognition | Socio-emotional | Gross/fine motor | |
| Bangladesh (Hossain et al.[53]) | 22 months (2 years) | BSID-III | BSID-III | Wolke Scales[d] | BSID-III[f] | Yes |
| Burkina Faso (Bouguen & Dillon[47]) | 0–6 years | CREDI, MELQO | CREDI, MELQO | CREDI | CREDI[f], MELQO[i] | Not reported |
| China (Wong et al.[54]) | 6–7 years | School readiness test for children in China (Ou 1990[103], Ou 2007[104]) | School readiness test for children in China (Ou 1990[103], Ou 2007[104]) | School readiness test for children in China (Ou 1990[103], Ou 2007[104])[e] | School readiness test for children in China (Ou 1990[103], Ou 2007[104])[g] | Yes |
| Ecuador (Paxson & Schady[57]) | 36–83 months (3–7 years) | TVIP | Woodcock-Johnson-Munoz[a] | BPI | Fine motor control pegboard exercise | Not reported |
| Ecuador (Fernald & Hidrobo[58]) | 12–35 months (1–3 years) | MacArthur IDHC-B | - | - | - | Adapted from a version validated in Mexico |
| Honduras (López Boo & Creamer[59]) | 9–67 months (0.75–5.5 years) | ASQ | ASQ | ASQ | ASQ[h] | Not reported |
| India (Weaver et al.[55]) | 3 years | - | ASQ | - | ASQ[h,i] | Not reported |
| Madagascar (Datta et al.[48]) | 0–6 years | MDAT | - | MDAT | MDAT[i] | Yes |
| Mexico (Fernald[60]) | 36–72 months (3–6 years) | TVIP | Revised Woodcock-Munoz | - | McCarthy Scales[h] | Not reported |
| Mexico (Ozer et al.[61]) | 56–68 months (4.5–5.5 years) | - | - | BPI | - | Adapted from a version validated in Mexico |
| Nepal (Levere et al.[56]) | 1–3 years | ASQ | ASQ | ASQ | ASQ[h,i] | Not reported |
| Nicaragua (Macours et al.[62]) | 0–83 months (0–7 years) | TVIP, DDST | DDST, McCarthy Scales[b], Woodcock-Johnson-Munoz[c] | DDST, BPI | DDST[h,i], McCarthy Scales, leg motor development | Not reported |
| Niger (Premand & Barry[49]) | 6–59 months (0.5–6 years) | - | Adapted BSID (Zeitlin & Barry[105]) | SDQ | - | Yes |
| Nigeria (Carneiro et al.[50]) | 4–6 years | ASQ | - | ASQ | ASQ[h] | Not reported |
| Tanzania (Sudfeld et al.[51]) | 18.9 months (1.5 years) | BSID-III | BSID-III | - | BSID-III[f] | Not reported |
| Uganda (Bjorvatn et al.[52]) | 4–6 years | IDELA | IDELA | IDELA | IDELA[f] | Not reported |

Direct Assessment Instruments: BSID-III Bayley Scales of Infant and Toddler Development, Third Edition, TVIP Test de Vocabulario en Imagenes Peabody, MDAT Malawi Development Assessment Test, DDST Denver Developmental Screening Test.
Caregiver-Report Instruments: CREDI Caregiver-Reported Early Development Instruments, MELQO Measuring Early Learning Quality and Outcomes, BPI Behavior Problems Index, IDHC-B Inventario del Desarrollo de Habilidades Comunicativas - Breve, ASQ Ages and Stages Questionnaire, SDQ Strengths and Difficulties Questionnaire.
-: not measured.
a Long-term memory, short-term memory, & visual integration
b Short-term memory
c Associative memory
d Emotional Tone
e Level of self-management
f Combined gross & fine motor
g Overall physical capacity & fine motor skills (hands)
h Gross motor
i Fine motor

**Table 3 | Cash-Plus Program Characteristics**

| Study (listed alphabetically by country) | Type of cash-plus program | Cash-plus component details | Duration of plus component | Cash transfer payment timing |
|---|---|---|---|---|
| Bangladesh (Hossain et al.[53]) | UCT+ | Psychosocial stimulation curriculum adapted from Reach Up, delivered by village health workers (VHWs) | 40–60 min sessions every fortnight for 1 year | Not reported |
| Burkina Faso (Bouguen & Dillon[47]) | UCT+ | T2: Livestock vouchers that could be exchanged for animals at designated fairs; value estimated at 11 poultry, or 3 goats or sheep T3: T2 + 2.5 kg of fortified flour per month per child aged 6–23 months, monthly supply of fortified flour per pregnant or lactating woman, and behavior change communication messages on the nutrition of pregnant women and young children | 3-4 months (lean season of June-September) for each of 2 years | Monthly, concurrent with the plus component |
| China (Wong et al.[54]) | CCT+ | Preschool tuition voucher for waiving preschool tuition up to 300 yuan per semester (tuition in rural areas typically ranges between 150 and 300 yuan per semester) | Beginning of semester for each of 2 semesters (1 year) | Final week of each of the 2 semesters, after verifying 80% attendance |
| India (Weaver et al.[55]) | UCT+ | Nutrition messaging delivered via flyers and a verbal message from a local early childhood care center worker at registration, and an automated call offering age-specific suggestions on nutritious foods to purchase at payment delivery | 1 year | Monthly, concurrent with the plus component |
| Madagascar (Datta et al.[48]) | UCT+ | Arm 1: Mother leaders (bimonthly meetings and home visits discussing health, nutrition, and ECD) Arm 2: Mother leaders and plan-making Arm 3: Mother leaders and self-affirmation | 18 months | Bimonthly |
| Nepal (Levere et al.[56]) | UCT+ | Local health worker-delivered information on best practices regarding nutrition and health for children below the age of 2; content included a focus on nutrition for mothers with offspring in utero, best practices during pregnancy, breastfeeding, care when sick, and supplemental feeding when older | 9 months | Monthly for the latter 5 months of the plus component |
| Nicaragua (Macours et al.[62]) | CCT+ | Group 1: Cash conditioned on school attendance for children ages 7–15 and nutrition, health, and education information package Group 2: Cash conditioned on school attendance for children ages 7–15 and vocational, labor market, and business skill training package Group 3: Cash conditioned on school attendance for children ages 7–15 and a lump-sum payment for a business development plan | 13 months (November-December) | Groups 1–3: Every 2 months Group 3: Lump-sums in May and September |
| Nigeria (Carneiro et al.[50]) | UCT+ | Low-intensity (posters, radio, preaching/Islamic school teachers, health talks, food demontrations, and prerecorded SMS/voice messages) and high-intensity (small group parenting sessions and 1-to-1 counseling) information channels with eight key messages covering practices of childcare and nutrition during pre-, peri-, and postnatal periods; messages also encourage mothers to increase their food intake during pregnancy and emphasize good hygiene and sanitation | 4 years | Monthly, concurrent with the plus component |
| Tanzania (Sudfeld et al.[51]) | CCT+ | Integrated health, nutrition, and responsive stimulation intervention in the home every 4–6 weeks; main maternal and child components included identification and referral for under-5 childhood illness per the Integrated Management of Childhood Illness, antenatal and postnatal counseling, danger signs identification, family planning, and emergency and routine referrals to facilities | 18 months | By visit, concurrent with plus component |
| Uganda (Bjorvatn et al.[52]) | UCT+ | Center-based childcare; typically preschool nurseries with lessons during the morning hours and (supervised) play or rest time in the afternoon | 1 year | Three installments, one per trimester (February, May, and September) |

## (a) Risk of bias using ROB2 for cluster RCTs (14 studies)

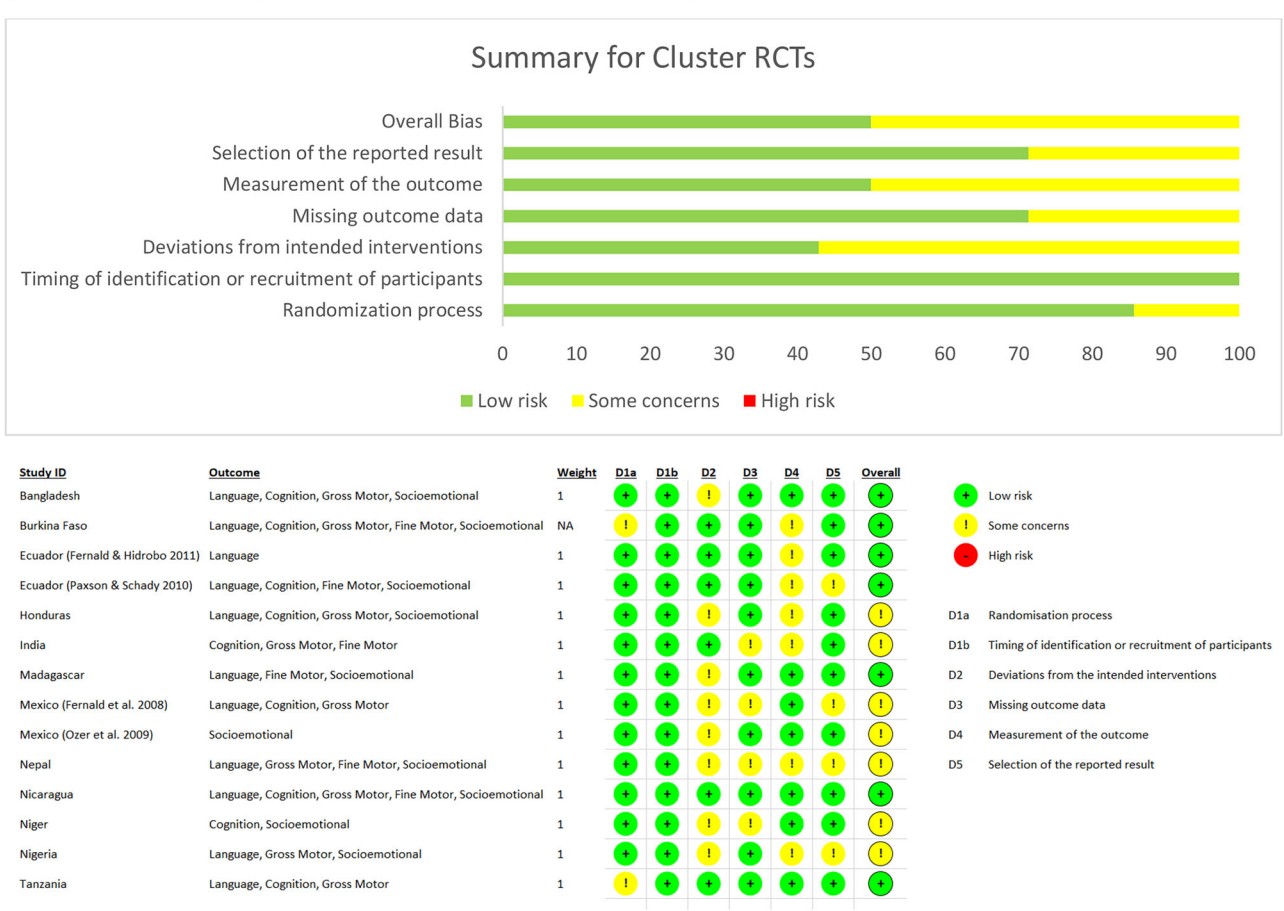

## (b) Risk of bias using ROB2 for individually-randomized RCTs (2 studies)

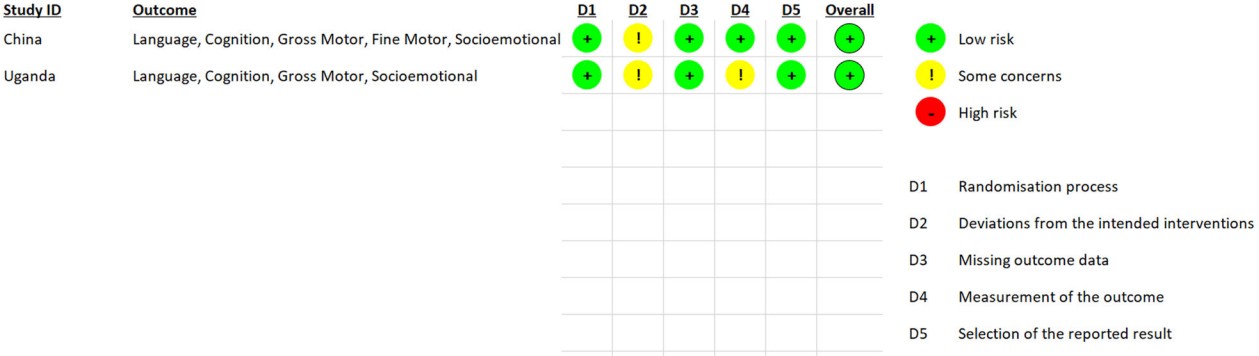

**Fig. 2 | Risk of Bias Assessment (RoB 2). a** Displays results of the risk of bias assessment for cluster RCTs (14 studies). **b** Displays results of the risk of bias assessment for individually-randomized RCTs (2 studies).

conditional cash transfer-plus programs, with an effect size of 0.13 SD (0.04, 0.22) (Fig. 6). Of the twelve studies of unconditional cash transfer or unconditional cash transfer-plus programs, there were no effects on socio-emotional outcomes in children, with one exception of a cash-plus program (Burkina Faso).

There were significant effects of cash transfer programs on gross motor skills of 0.07 SD (0.03, 0.11) (Fig. 7) but no significant effects on fine motor skill development (Fig. 8). One conditional cash transfer-plus program (Tanzania, 0.18 SD) and two unconditional cash transfer-plus programs (Uganda, 0.19 SD and Bangladesh, 0.22 SD) achieved the largest effect sizes on gross motor skills. The plus components of these three programs were

livestock, fortified flour, and nutrition information; psychosocial stimulation; and community health worker-delivered health, nutrition, and responsive stimulation, respectively. Only two unconditional-plus programs showed effects on fine motor skills, including those in India and Burkina Faso.

### Role of transfer size

We next examined the associations between program transfer generosity and effect sizes in cognition, language, socio-emotional, gross motor, and fine motor outcomes (Fig. 9). Visual inspection reveals a pattern of increasing transfer generosity associated with increasing effect sizes for

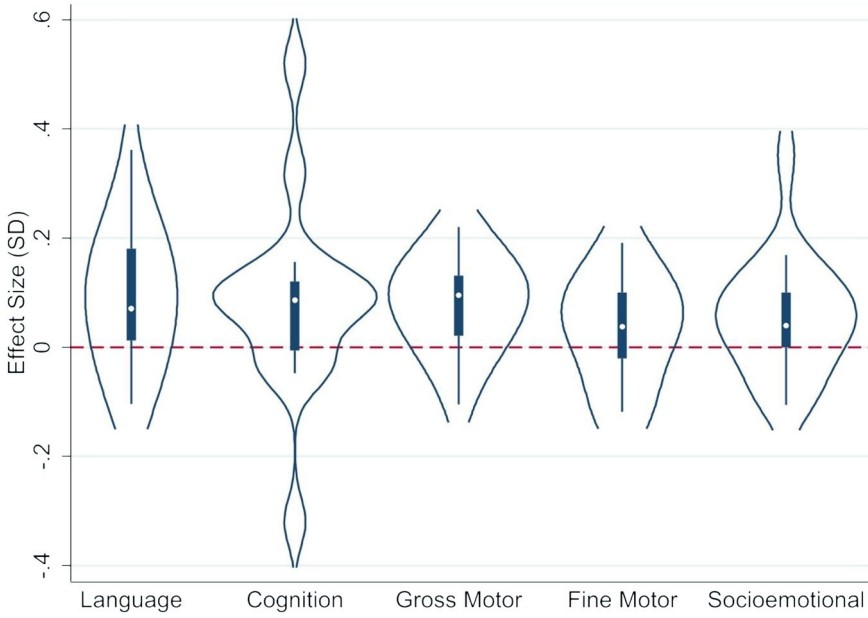

**Fig. 3 | Violin plots of the distribution of effect sizes by domain.** Language: *N* = 18 effect sizes (13 studies), Cognition: *N* = 16 effect sizes (11 studies), Gross Motor: *N* = 16 effect sizes (11 studies), Fine Motor: *N* = 9 effect sizes (7 studies), Socio-emotional: *N* = 17 effect sizes (12 studies).

socio-emotional outcomes, and a similar but less obvious trend for cognition. There does not appear to be this same trend for language, gross motor, or fine motor skills. We were not able to conduct statistical tests of these trends due to small sample sizes.

### Relevance of conditionality
Participating in a cash transfer programs with a component of conditionality was more likely to be associated with higher scores in language, socio-emotional, and gross motor outcomes, whereas results were mixed for cognition and fine motor (Fig. 10). In tests of heterogeneity by world region, there was a consistent trend that effect sizes hovered around zero for studies conducted in Africa and Asia for all outcomes (Fig. 11). In contrast, studies conducted in Latin America demonstrated significant effects in all domains.

## Discussion
### Key findings
After an average follow-up time of 3.6 years, cash transfers had small but statistically significant positive effects on cognitive, language, and gross motor skills among recipients, whereas effects on socio-emotional outcomes and fine motor skills were not significant. Conditional cash transfer programs had slightly larger effect sizes than unconditional programs in terms of promoting children's development, with the greatest benefits for cognitive, language, and socio-emotional skills and less for fine motor and gross motor skills. There was a positive association between transfer generosity (as a percent of income) and cognitive and socio-emotional outcomes. The "plus" components (such as parenting support or nutrition education) added more value to unconditional than conditional programs. When looking at effects on child cognition, cash in a program bundled with sessions engaging parents and children in play activities outperformed cash bundled with more passive communication approaches. Studies conducted in Latin America appeared to demonstrate more significant effects across outcomes, whereas those from Africa and Asia showed more heterogeneity.

### Role of conditionality
Overall, conditional cash transfer programs were more effective at improving child development compared with unconditional cash transfer programs. This finding suggests that children's development may improve as a consequence of several well-established pathways unrelated to the cash transfer itself. For example, there are clear benefits of conditional cash transfer program participation on antenatal care attendance, especially

among poorer populations[63], having a skilled birth attendant[64–67] and births at clinics[68–71], all of which could affect child development. Several conditional cash transfer programs, not included in this review, have focused on improving child health and are conditional on parents taking the children to health visits[64,72–80]. Children in these programs have a higher likelihood of being taken to health facilities for growth monitoring or preventive care, in contrast to children in unconditional programs, which have not found an increase in child health behaviors[57,81,82]. The quality and quantity of household food is another pathway that could explain differences in conditional and unconditional program effectiveness, though recent reviews suggest that both types of programs are associated with improvements in dietary diversity[83,84]. The larger impacts of conditional relative to unconditional programs on cognitive, language, and socio-emotional skills may reflect the higher developmental sensitivity of these domains to enriched caregiving, schooling, and health-seeking behaviors, behaviors that conditional programs explicitly incentivize. In contrast, fine and gross motor skills show smaller SES-related baseline disparities and rely more heavily on maturational processes, which may limit the potential for differential impacts between transfer modalities.

These results are consistent with a recent meta-analysis of only unconditional cash transfer programs on a range of outcome measures, which found that in the programs that signal that the cash is intended to benefit children, there were greater effects on child health outcomes[84]. Conditional cash transfer programs typically embody a top-down framework where individuals or organizations decide what is best for poor children and provide incentives to their parents to achieve these objectives. Conversely, unconditional programs operate on the premise that once a budget constraint is relaxed, parents are in a better position to make appropriate decisions regarding their child's human capital. The existing literature suggests that recipients have often felt that the cash transfer amount was not sufficient given their needs, and that cash alone is not enough to change behavior and should be supplemented with additional support[85], although there are very few studies exploring these questions. Taken together, these findings suggest that parents are better able to provide optimal support for child development when services besides cash are provided. Future research should prioritize multi-arm designs that disentangle the income effect from the enforcement effect, helping policymakers determine whether the administrative burden of verifying conditions is necessary to achieve developmental gains. In addition, while "cash-plus" programs generally outperformed "cash-only" interventions, a critical

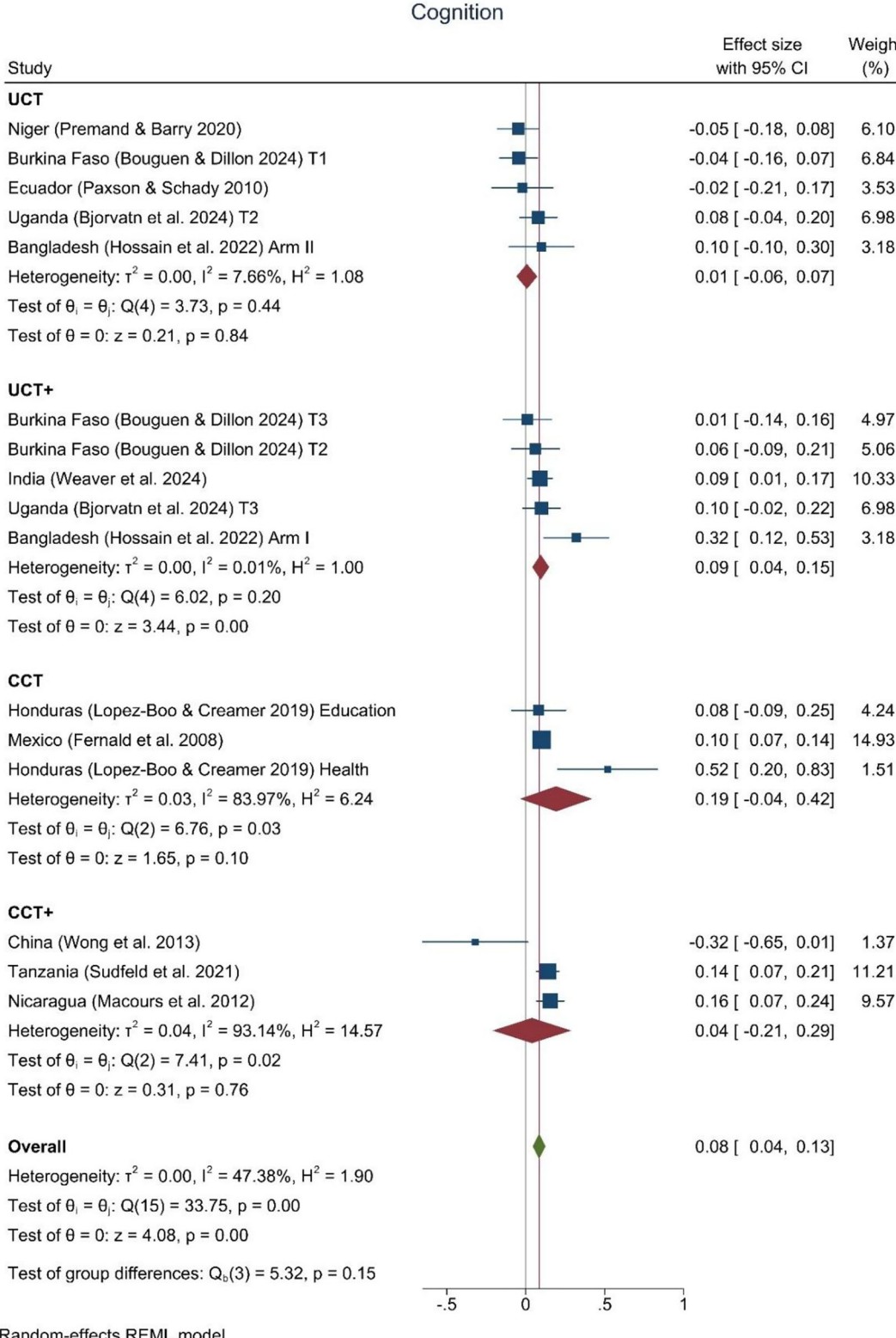

**Fig. 4 | Cognition effect sizes by type of program.** UCT: $N = 5$ effect sizes (5 studies), UCT+: $N = 5$ effect sizes (4 studies), CCT: $N = 3$ effect sizes (2 studies), CCT+: $N = 3$ effect sizes (3 studies), Overall: $N = 16$ effect sizes (11 studies).

outstanding question is whether the cash and services act synergistically, or if the "plus" components drive the impact independently of the financial transfer. This question could only be answered using 2×2 factorial designs to determine when cash is a necessary driver, and when supply-side investments alone would be sufficient.

Given that our findings suggest greater benefits to child development from involvement in conditional cash transfer programs compared with unconditional programs, a key question is whether and how to require conditions for the transfer. Theoretically, conditional and unconditional programs differ because conditional programs are designed with the

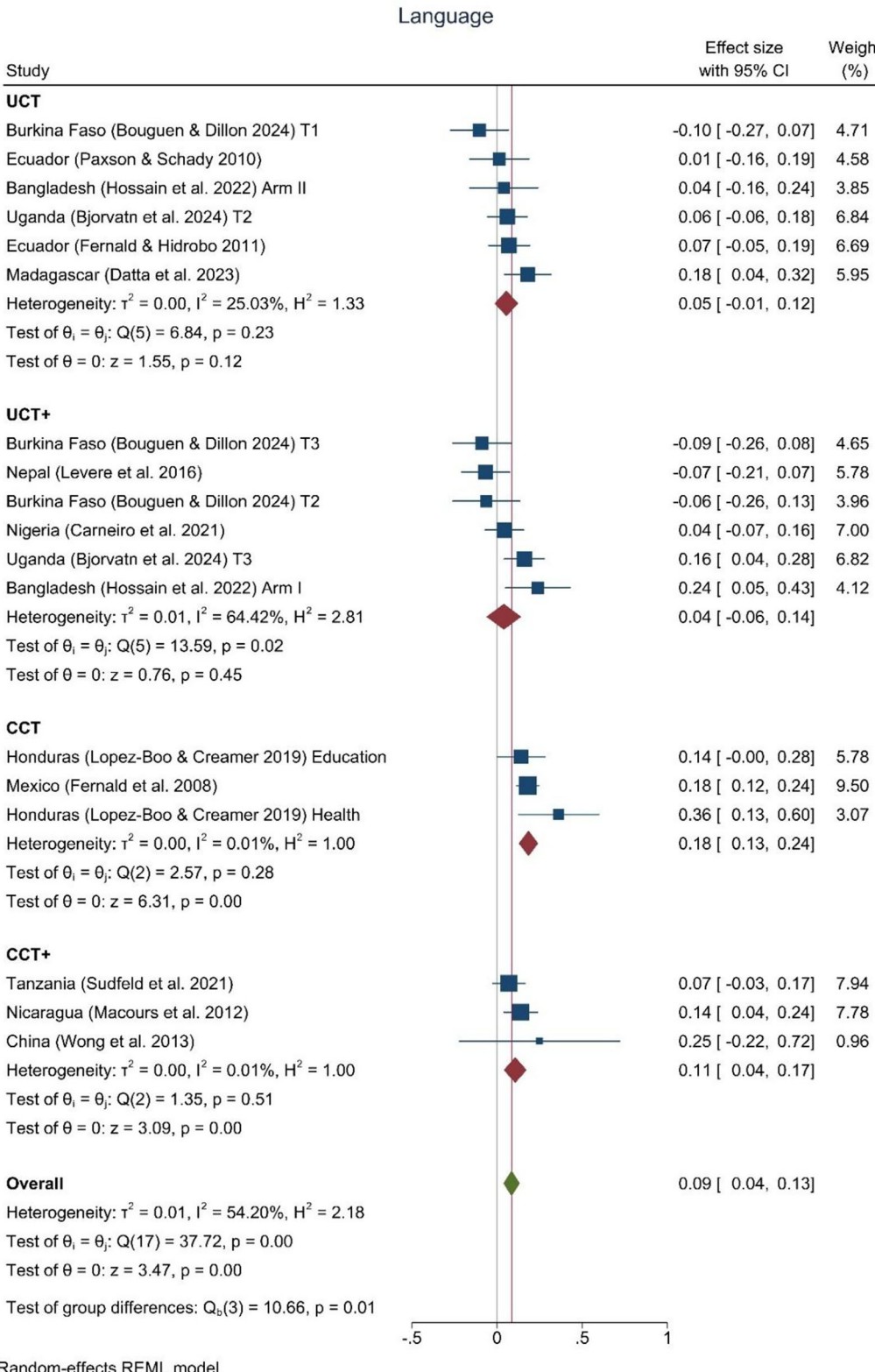

**Fig. 5 | Language effect sizes by type of program.** UCT: $N = 6$ effect sizes (6 studies), UCT+: $N = 6$ effect sizes (5 studies), CCT: $N = 3$ effect sizes (2 studies), CCT+: $N = 3$ effect sizes (3 studies), Overall: $N = 18$ effect sizes (13 studies).

assumption that there are certain behaviors, such as healthcare visits or nutrition supplementation, that are optimal for children, and then the programs provide incentives for parents to participate in those programs. In contrast, unconditional programs assume that when a family's income is raised, the parents will make decisions that optimize outcomes for their children. Proponents of conditionality argue that caregivers may have incomplete or incorrect information about the benefits of health or education investments, or that individuals may suffer from myopia or procrastination, which results in a lack of alignment of short- and long-term goals, and that there may be intra-household conflicts of interest between

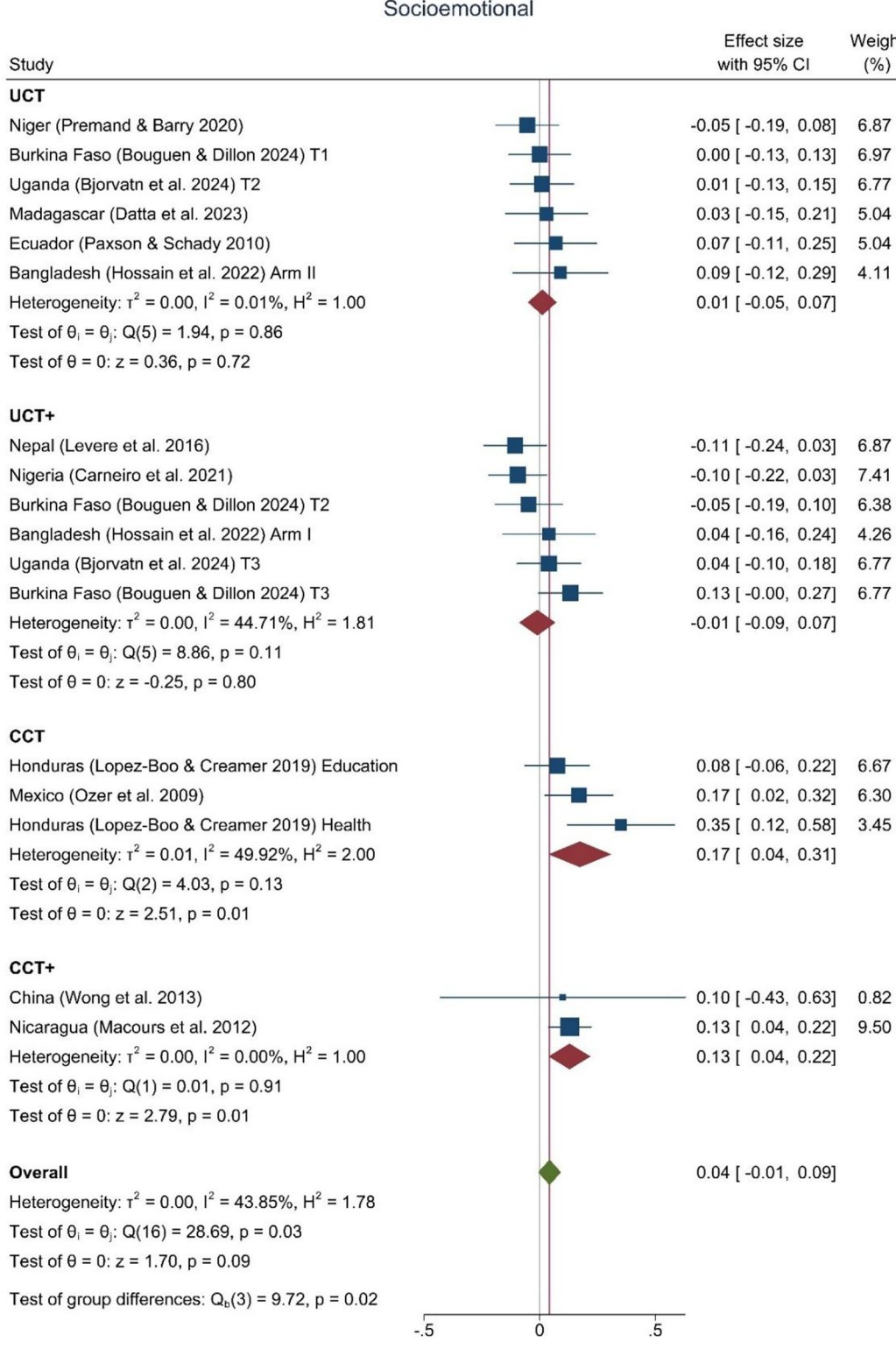

**Fig. 6 | Socio-emotional effect sizes by type of program.** UCT: $N = 6$ effect sizes (6 studies), UCT+: $N = 6$ effect sizes (5 studies), CCT: $N = 3$ effect sizes (2 studies), CCT+: $N = 2$ effect sizes (2 studies), Overall: $N = 17$ effect sizes (12 studies).

parents and children[86]. Although many studies show improvements in food consumption and health service use as a result of required conditions, many factors could limit the effectiveness of conditions in improving child development outcomes, especially in contexts including environmental risks like contaminated water and malaria[87,88].

There are clear ethical considerations to weigh when determining what the conditions will be for a given program[89]. For example, a growing literature addresses the issue of the ethics of enforcing conditions, which may exclude the poorest households if the transfer does not cover the costs of compliance, such as the cost of getting to the healthcare center

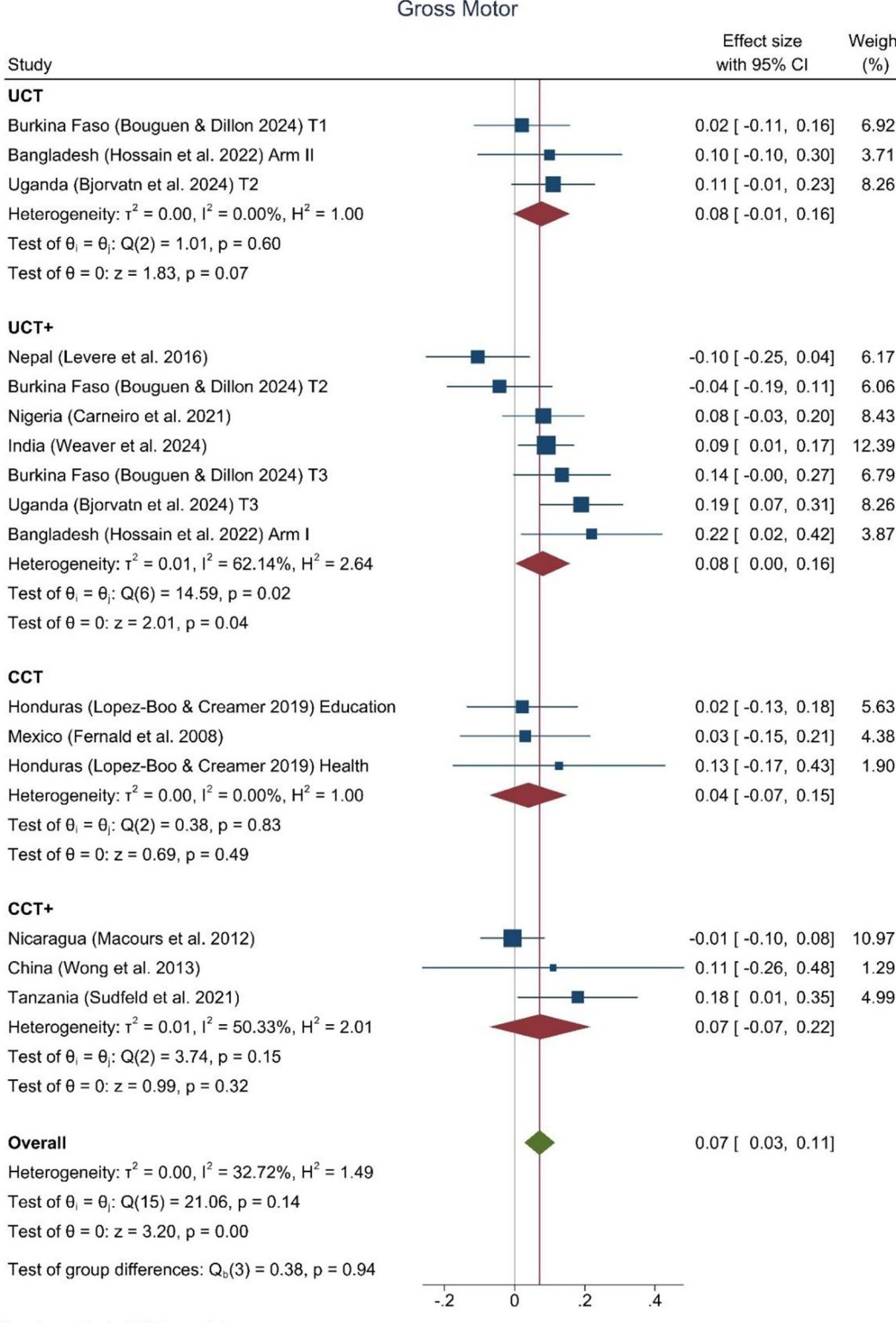

**Fig. 7 | Gross motor effect sizes by type of program.** UCT: $N = 3$ effect sizes (3 studies), UCT+: $N = 7$ effect sizes (6 studies), CCT: $N = 3$ effect sizes (2 studies), CCT+: $N = 3$ effect sizes (3 studies), Overall: $N = 16$ effect sizes (11 studies).

for a mandatory clinic visit[90]. There may also be exacerbation of existing injustices when additional burdens are imposed on already vulnerable and marginalized populations, and it may also be unethical to require participants to attend schools or health clinics that are of very poor quality.

**"Plus" component**

Our heterogeneity analysis compared cash transfer programs with cash transfer "plus" other components (such as parenting support or nutrition education) and found that the "plus" components added more value to unconditional than the conditional programs. A key question has to do with

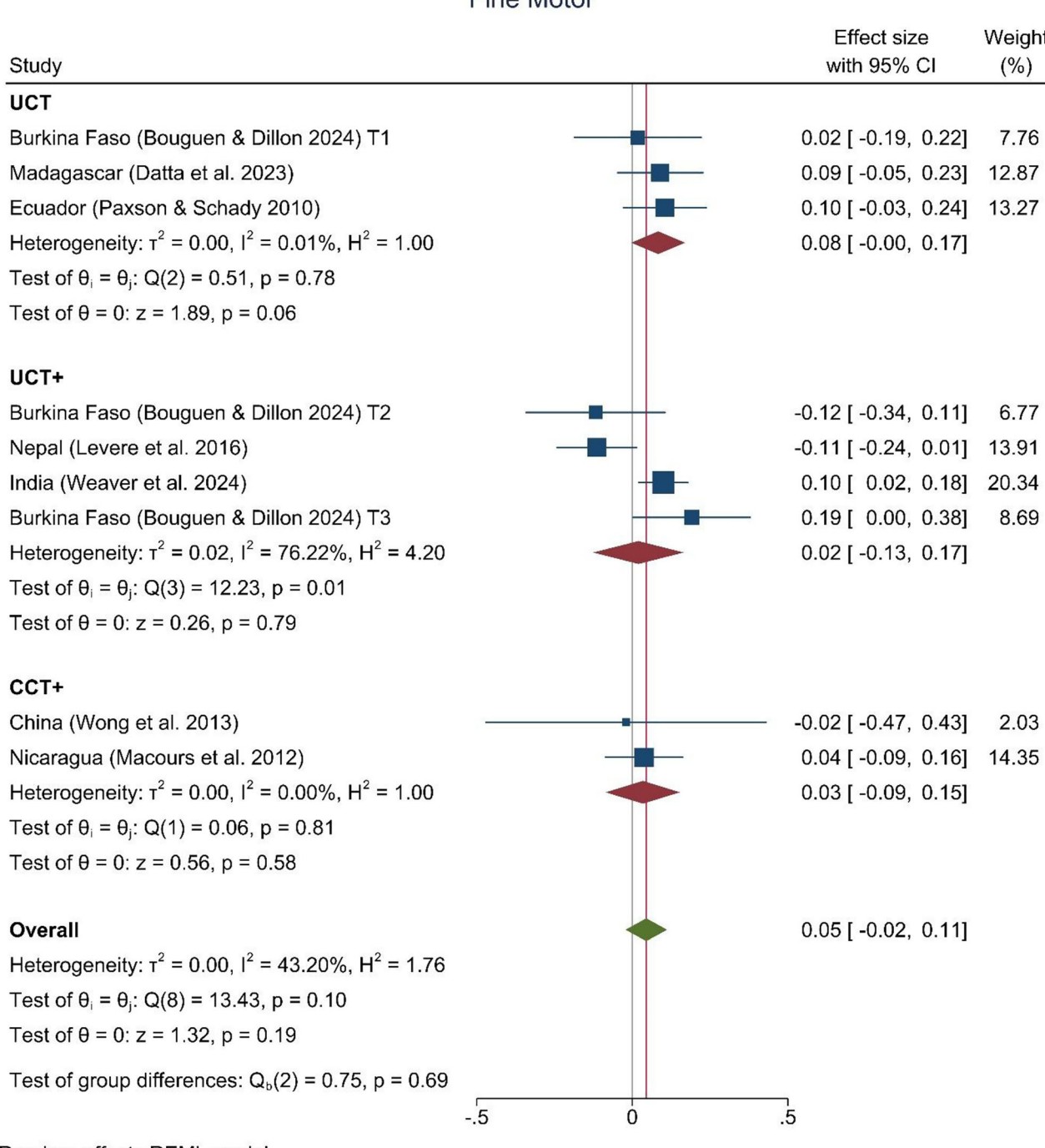

**Fig. 8 | Fine motor effect sizes by type of program.** UCT: $N = 3$ effect sizes (3 studies), UCT+: $N = 4$ effect sizes (3 studies), CCT+: $N = 2$ effect sizes (2 studies), Overall: $N = 9$ effect sizes (7 studies).

the use of "plus" components to cash transfers, which are designed to be complementary components to improve effectiveness by targeting mediating outcomes or availability of supplies or services. Cash bundled with sessions engaging parents and children in play activities outperformed cash bundled with more passive communication approaches, as well as cash-only programs, when looking at effects on child cognition. These findings may reflect the specific conditions that the conditional and conditional-plus imposed, along with the intensity of the "plus" components. For instance, in Nicaragua and Honduras, transfers were conditional on children attending preventive health checkups and being enrolled in school, and in China, the

"plus" component was a preschool tuition voucher. Meanwhile, most of the unconditional-plus programs had relatively light-touch "plus" components (information packages, messaging, plan-making, and affirmations), with one exception in one arm of the trial in Bangladesh, which delivered psychosocial stimulation. This unconditional-plus arm had the second-largest cognitive effect size of all studies, in line with impacts observed in other studies of psychosocial stimulation interventions in LMICs. It is perhaps unsurprising that cash bundled with sessions engaging parents and children in play activities outperformed cash bundled with more passive communication approaches, as well as cash-only programs on child cognition. One

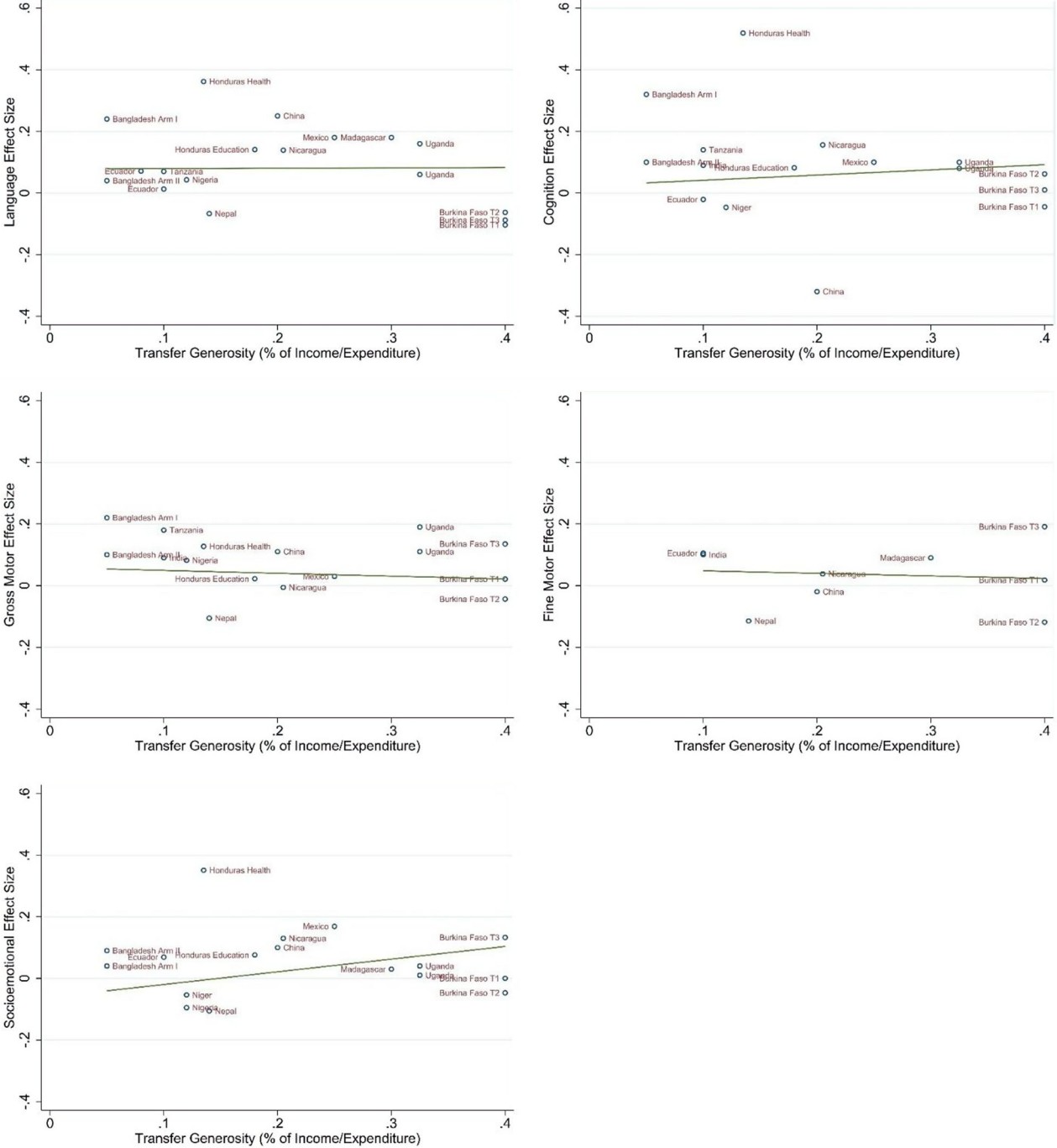

**Fig. 9 | Association between transfer generosity and cognition, language, socio-emotional, gross motor, and fine motor effect sizes, weighted by sample size.** Language: $N = 18$ effect sizes (13 studies), Cognition: $N = 16$ effect sizes (11 studies), Gross Motor: $N = 16$ effect sizes (11 studies), Fine Motor: $N = 9$ effect sizes (7 studies), Socio-emotional: $N = 17$ effect sizes (12 studies).

unconditional-plus program (Nepal) produced significant impacts on cognition, yet only achieved a small effect size, likely because the plus component was light-touch messaging (automated calls), encouraging the recipient to spend the cash on nutritious food for the mother and child.

A recent systematic review and meta-analysis examined the specific question of whether cash-plus interventions were more effective than cash alone in terms of improving health and well-being, and found that cash-plus was more effective for height-for-age, but not weight-for-height or for cognitive development[41]. There was some suggestive evidence that cash-plus-healthcare may contribute to reductions in mortality, and cash-plus-food was more effective in preventing acute malnutrition in crisis contexts, but there are

clearly many more questions to answer in this domain. A government-led program in rural Mexico layered group-based parenting support onto the existing conditional cash transfer program, and showed benefits to early child development outcomes, particularly among indigenous children[91], reinforcing the notion that optimal child development outcomes can be achieved by providing cash transfers paired with additional forms of support for parents as well as social-sector investment such as early childhood education.

Only two of our studies included a "plus-only" arm that could be directly compared to a cash-plus arm. In Tanzania, the cash-plus arm had impacts on cognition, language, and motor skills, compared to only cognition for the plus-only arm. In Uganda, the impacts in the cash-plus arm

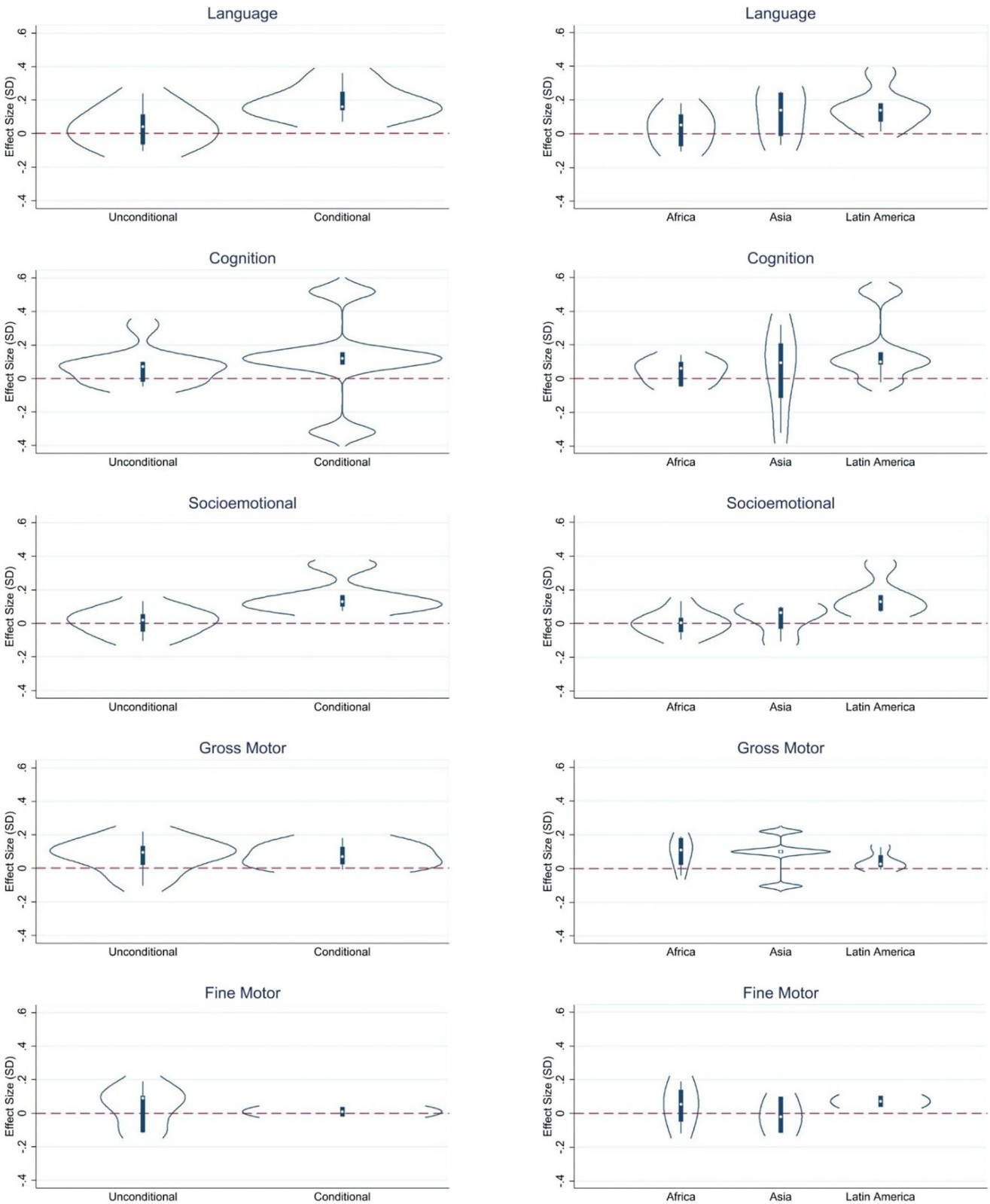

**Fig. 10 | Distribution of effect sizes by domain and program conditionality.** Language: $N = 18$ effect sizes (13 studies), Cognition: $N = 16$ effect sizes (11 studies), Socio-emotional: $N = 17$ effect sizes (12 studies), Gross Motor: $N = 16$ effect sizes (11 studies), Fine Motor: $N = 9$ effect sizes (7 studies).

**Fig. 11 | Distribution of effect sizes by domain and geographic region.** Language: $N = 18$ effect sizes (13 studies), Cognition: $N = 16$ effect sizes (11 studies), Socio-emotional: $N = 17$ effect sizes (12 studies), Gross Motor: $N = 16$ effect sizes (11 studies), Fine Motor: $N = 9$ effect sizes (7 studies).

were no different from those in the plus-only arm across all domains. It seems plausible that the value-added of cash depends on the outcome domain, the "plus" component, and the context. Specifically, effectiveness relies on whether the cash can easily "buy" some of the inputs provided by the "plus" component or otherwise substitute for developmental inputs in a given setting, and the extent to which those specific inputs drive results in different outcome domains. Future work should address the extent to which cash generates distinct benefits from potentially more intensive or expensive "plus" components, and the conditions under which it acts synergistically with plus components to increase developmental inputs.

## Possible mechanisms

To better understand mechanisms of program effects, we examined compliance with conditions and program uptake in seven of the 16 studies that explicitly measured these metrics. First, administrative fidelity regarding the cash transfer itself was consistently high: receipt rates ranged between 90% and 99% in India, Niger, Nigeria, and Tanzania, regardless of conditionality. However, uptake of the "plus" components and adherence to behavioral conditions showed significant heterogeneity. For example, in Burkina Faso, while 95% of households received the cash, uptake of the "plus" components was lower, with only 76% receiving the intended animal assets and 65% the enriched flour. Similarly, compliance with behavioral conditions varied by domain: while the Tanzania program achieved 99.5% compliance with health clinic visits, the China program reported only 74% compliance with preschool attendance requirements. Finally, we found evidence of implementation failure regarding conditionality: the Ecuador program, despite being designed as a conditional cash transfer, failed to enforce its health and education requirements, effectively operating as an unconditional transfer program. This variation in compliance, particularly regarding the "plus" components and inconsistent enforcement of conditions, may help explain why outcome sizes differed across contexts even when study designs appeared similar.

Beyond compliance, another pathway through which child development could be affected by cash transfer program participation is through parental mental health[92]. Effects of parental depression can be transmitted inter-generationally through maternal stress, epigenetic effects, impaired fetal growth, altered maternal behaviors including mother-child interactions, and ultimately child growth, health, and development[26]. A recent systematic review and meta-analysis reported that cash transfers significantly reduced depression and anxiety among adults and adolescents, but that improvements were not often sustained at 2–9 years after the end of the program; effects on stress were insignificant[93]. However, another meta-analysis focusing on children and young people found null effects of cash transfers on depressive symptoms[94]. Among the programs included in our meta-analysis, there were effects on maternal depression in Mexico[95], but not in Ecuador[81], or Nicaragua[96]. In India, a separate, quasi-experimental evaluation of a one-time cash transfer program demonstrated improved mental health for mothers, but did not measure early child development[97]. Additional research is needed to understand how conditional and unconditional cash transfer programs differentially affect mental health and other outcomes for recipient parents, such as stress.

Intra-household bargaining research papers indicate that resources under the mother's control have a stronger positive impact on a child's health and schooling than resources controlled by the father[98,99]. However, almost all current cash transfer programs give resources to the mother, including all of the papers included here, so we cannot disentangle how much of the impact is due to the recipient's gender. A recent meta-analysis of unconditional programs with a broad range of outcomes beyond child development concluded that effects on consumption and income were greater for those targeted to women[100]. Future research could explore whether a recipient's gender might affect outcomes differently for conditional as opposed to unconditional cash transfers.

## Limitations

A key limitation of our analysis was the small number of papers that fit our inclusion criteria, which made it difficult to examine heterogeneity statistically due to small sample sizes. Given that there are hundreds of cash transfer programs worldwide, we are clearly examining a tiny sample of programs, which are likely to have been more controlled than a larger, scaled-up program. While we adhered to a rigorous protocol that included gray literature to minimize publication bias, we acknowledge that excluding quasi-experimental studies reduces our sample size. Consequently, our comparisons of effect sizes by program conditionality and region are descriptive and exploratory in nature, and our results represent a conservative estimate of causal impact derived strictly from experimental evidence.

We were able to visually examine violin plots to examine findings by program conditionality and region of study, but future research could address this question more empirically. We are also limited in our ability to examine heterogeneity by region because most conditional cash transfer programs have been administered in Latin America, whereas unconditional transfers are largely concentrated in Asia and Africa. The observed advantage of programs with conditions may reflect the higher quality of supply-side services (schools and clinics) in Latin American middle-income contexts rather than the conditionality itself.

Another limitation of our analysis is that we are examining all conditional cash transfer programs together, regardless of the type of condition, although the conditions are largely focused on improving child health and development outcomes. We were unable to find studies with multiple treatment arms, which explicitly tested both unconditional and conditional transfers with ECD outcomes, and this is a promising area for future research. Finally, we could not generate funnel plots to test publication bias because of the small number of papers, which is another limitation.

## Conclusions

In spite of these limitations, our analysis can add to the conversation about poverty alleviation efforts, given the widespread popularity and use of cash transfers as instruments for improving economic outcomes. It is already well established that many factors shape the effectiveness of cash transfer programs for improving health outcomes, including approach to targeting, benefit adequacy, and baseline levels of poverty among recipients[101]. Furthermore, a recent analysis of 17 programs in Latin America concluded that cash transfer programs could be more effective at reducing poverty and inequity if there was a greater effort to increase coverage among the poor, as well as more frequent recertification of eligibility in order to create more efficient systems[102]. Our meta-analysis adds to this work and suggests that cash transfers can be more effective for early child development if the programs are designed to be conditional, with required investments in health, nutrition or education, or when they are designed as "cash-plus" rather than just cash alone. For policymakers specifically interested in improving early childhood development, our findings point to the importance of bundling cash with investments in nutrition, health, and education. This evidence can be used to guide future decisions about cash transfer programs specifically focused on improving early child development, which should then be adapted based on context, policy goals, administrative capacity, and current investments.

Future decisions regarding cash transfers must balance specific policy goals with administrative and contextual realities. While cash alone reduces poverty, our findings suggest that the policy goals of achieving gains in cognitive and language development require conditional cash transfer programs or "cash-plus" models that explicitly integrate child-focused components. However, the choice of design must align with administrative capacity and local context; in settings where monitoring infrastructure is weak or where concurrent investments in the quality of health and education services are lacking, enforcing strict conditions may be operationally unfeasible or ineffective. Consequently, where supply-side services are inadequate, policymakers could prioritize "cash-plus" approaches that bundle financial support with direct interventions, such as psychosocial stimulation or nutritional counseling, thereby using the transfer as a platform to deliver multi-sectoral benefits without the exclusion risks of rigid conditionality.

## Data availability

Data extracted from studies included in the meta-analysis are available at https://github.com/eleanortsai/Cash-Transfers-ECD.

## Code availability

R script and code to generate figures are available at https://github.com/eleanortsai/Cash-Transfers-ECD.

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

## Acknowledgements

The authors received no specific funding for this work. We acknowledge the contributions of Melissa Hidrobo and Damian de Walque, who contributed to previous reviews of this topic, and James Manley who contributed to a broader recent review chapter[42].

## Author contributions

L.F. and P.G. conceived the background research, idea, and concept. E.T. conducted the literature review and assembled and structured the source data for the meta-analysis with support from L.F. and P.G. All authors contributed to data analysis and interpretation of results. L.F. wrote the manuscript with input from co-authors and made all revisions of the manuscript. All authors critically reviewed and approved the final version of the manuscript. E.T. created all figures and assembled all source data into a repository.

## Competing interests

The authors declare no competing interests.
