## [Transparent Peer Review file · Communications Psychology]

A Systematic Review and Meta-analysis of Studies testing Effects of Cash Transfers on Child Cognitive, Language and Socio-Emotional Development in Low- or Middle Income Countries

Corresponding Author: Professor Lia Fernald

This manuscript has been previously reviewed at another journal. The manuscript was considered suitable for publication without further review at Communications Psychology.

Version 0:

Decision Letter:

Dear Professor Fernald,

Your manuscript titled "Effects of Cash Transfers on Child Cognitive, Language, and Socio-emotional Development: A Systematic Review and Meta-analysis of Studies in Low- or Middle-Income Countries" has now been seen by Reviewers #1 and #2. In light of their advice I am delighted to say that we are happy, in principle, to publish a suitably revised version in Communications Psychology.

We therefore invite you to revise your paper one last time to address the remaining concerns of our reviewers and a list of editorial requests. At the same time we ask that you edit your manuscript to comply with our format requirements and to maximise the accessibility and therefore the impact of your work.

EDITORIAL REQUESTS:

SUBMISSION INFORMATION:

OPEN ACCESS:

* **DATA AVAILABILITY:**

Link Redacted

Best regards,

Marike

Marike Schiffer, PhD
Chief Editor
Communications Psychology

REVIEWERS' COMMENTS:

Reviewer #2 (Remarks to the Author):

The authors have provided detailed and thoughtful responses to the issues I initially raised. I have no further comments.
